# Dynamic population-based meta-learning for multi-agent communication with natural language

**Abhinav Gupta**\*
MILA
abhinavg@nyu.edu

**Marc Lanctot**
DeepMind
lanctot@deepmind.com

**Angeliki Lazaridou**
DeepMind
angeliki@deepmind.com

## Abstract

In this work, our goal is to train agents that can coordinate with seen, unseen as well as human partners in a multi-agent communication environment involving natural language. Previous work using a single set of agents has shown great progress in generalizing to known partners, however it struggles when coordinating with unfamiliar agents. To mitigate that, recent work explored the use of population-based approaches, where multiple agents interact with each other with the goal of learning more generic protocols. These methods, while able to result in good coordination between unseen partners, still only achieve so in cases of simple languages, thus failing to adapt to human partners using natural language. We attribute this to the use of *static* populations and instead propose a *dynamic* population-based meta-learning approach that builds such a population in an iterative manner. We perform a holistic evaluation of our method on two different referential games, and show that our agents outperform all prior work when communicating with seen partners and humans. Furthermore, we analyze the natural language generation skills of our agents, where we find that our agents also outperform strong baselines. Finally, we test the robustness of our agents when communicating with out-of-population agents and carefully test the importance of each component of our method through ablation studies.

## 1 Introduction

Humans excel at large-group coordination, with natural language playing a key role in their problem solving ability [49, 8]. Inspired by these findings, in this work our goal is to endow artificial agents with communication skills in natural language, which can in turn allow them to coordinate effectively in three key situations when playing referential games (i) with agents they have been paired with during their training (ii) with new agents they have not been paired with before and (iii) with humans.

Our starting point is recent work in the emergent communication literature, which proposes to learn protocols "from scratch", allowing like these agents that have been *trained together* (and for a large number of iterations) to form conventions [47, 24, 12, 19]. While such communication approaches result in agents achieving higher rewards at test time *when paired with their training partners* [21, 14, 23], there is growing evidence that these agents do not acquire communication skills with general properties [31, 1, 3, 18] but rather learn to form communication protocols that are based solely on idiosyncratic conventions.

Interestingly, these type of communication pathologies are also observed in natural languages, where they are linked to the size of a linguistic community [42, 54, 50, 35], i.e. the smaller the

---

\*Work partially done during an internship at DeepMind.

35th Conference on Neural Information Processing Systems (NeurIPS 2021).

community the more complex the languages get. This has recently inspired the use of population-based approaches to multi-agent communication in which agents within a population interact with one another [48, 5, 11, 13]. One such approach is L2C [32] which uses meta-learning approaches like MAML [10] combined with a static population of agents. Indeed, L2C results in communication protocols with some primitive forms of compositionality, allowing agents to successfully communicate with unseen partners. Nevertheless, even in this case, the induced protocols fail to adapt to natural language, leaving open questions with regard to coordination with humans. We hypothesize that a potential bottleneck of this and related approaches lies in the use of static populations. Intuitively, the initial diversity induced by different random initializations of the population gradually vanishes when given enough capacity and training iterations, thus allowing all agents in the population to co-adapt to each other and converge to the same ad-hoc conventions. Simply put, it is challenging to maintain diversity in static populations, which is what drives the learning of generalizable protocols.

With this in mind, we propose to address this issue by dynamically building the population (§3), to gradually induce constructive diversity within the population, so as to enable population-based algorithms (like meta-learning) to train agents that can then generalize to seen and unseen partners as well as humans. Since communication with humans is one of our goals, grounding to natural language is of paramount importance. Consequently, we want the agents to coordinate with humans using limited amount of human data while not drifting away from human behavior. In order to maximize efficiency and minimize the use task-specific language data, we combine two learning signals [25, 33], each bootstrapping the other (§2). First, maximizing task rewards allows the agents to learn to communicate with one another in a purely self-play manner, but using emergent protocols not grounding in natural language. Hence, combining this signal with supervision from a *limited* dataset of task-specific language data, allows grounding of the emergent protocols in natural language. All in all, our proposed approaches consists of three different phases (i.e., interactive reward-based learning, supervised learning, and meta-learning), each imposing an inductive bias.

We present extensive experiments in two referential game environments (an image-based and a text-based defined in §4) using natural language communication and compare our model against previous approaches, strong baselines and ablations of our method. Moreover, we follow a holistic approach to evaluating multi-agent communication, reporting results on: (a) task performance measured by referential accuracy in (§4.1) (b) the agents' natural language generation skills by computing BLEU score with language data (§4.2) (c) human evaluation of both agents in (§4.3) (d) cross-play evaluation with unseen partners in (§4.5), and (e) robustness against implict bias in the dataset (§4.6).

## 2  Learning in multi-agent games with natural language: The case of referential games

**Referential Games**  Referential games are a type of Lewis Signaling games [28] that have been used in human as well as artificial language experiments [15, 23, 9, 27, 29, 13]. This fully cooperative game consists of two players, a speaker $\mathcal{S}$ and a listener $\mathcal{L}$ who interact with each other to solve a common task. In our setup, both agents are parameterized using deep neural networks, where the speaker's and listener's parameters are denoted by $\theta$ and $\phi$ respectively. The speaker gets as input a target object $t$, encodes it in its own embedding space and sends a discrete message $m = \mathcal{S}(t)$ to the listener. The listener

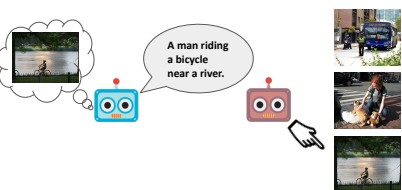

Figure 1: Referential Game with images.

receives two inputs, the message $m$ and a set of distractor objects $D$ ($|D| = K$) and the target object $t$. The listener embeds both of these into a shared vector representation to compute a similarity score between the message and the objects and outputs a prediction $t'$ about the target object. Both agents are given a reward if the listener is able to correctly predict the target object. We denote the maximum length of the message $m$ with $l$ and the vocabulary set with $V$. In our work, we use two different types of objects: images and sentences as described in §4.

**Interactive Learning**  The most common approach to train agents to solve the main communicative task in the multi-agent communication framework is by using interactive (reward-driven) learning [9, 37, 15, 29]. The reward function $r$ for both agents is the same and given by: $r = 1$ if $t =$

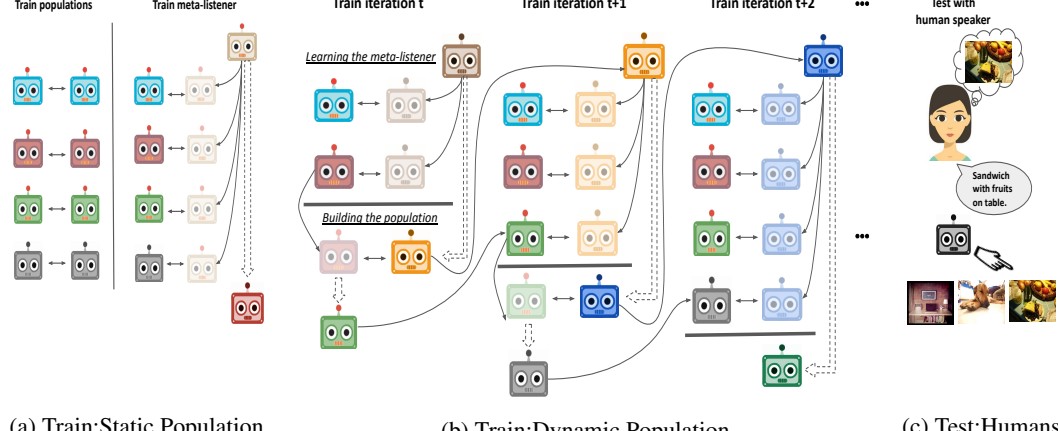

|(a) Train:Static Population | (b) Train:Dynamic Population | (c) Test:Humans |

Figure 2: Meta-learning a listener using two types of population. In Fig 2a, a fixed set of speakers and listeners are trained using self-play. Then a meta-listener is trained by interacting with the fixed population of speakers. In Fig 2b, the meta-listener is trained using the current population of speakers at time $t$ in top-phase. Speakers are added to this population iteratively by training the latest speaker (obtained at $t-1$) with the updated meta-listener (denoted by dashed arrow) in the bottom-phase. In Fig 2c, the final trained meta-listener plays with human speakers during test time.

$t'$ and $-0.1$ otherwise. We optimize the agents' parameters using reinforcement learning, and specifically, using policy gradients (REINFORCE [53]), similar to [9, 25]. The listener is additionally optimized using a supervised learning (cross-entropy) loss since we know the ground truth label (which is the target object). The corresponding interactive loss functions for the speaker ($\mathcal{J}_{\mathcal{S}}^{\texttt{int}}$) and the listener ($\mathcal{J}_{\mathcal{L}}^{\texttt{int}}$) are given by:

$$\mathcal{J}_{\mathcal{S}}^{\texttt{int}}(t;\theta) = -\frac{r}{l}\sum_{j=1}^{l}\log p(m_j|m_{<j},t;\theta) + \lambda_{hs}H_{\mathcal{S}}(\theta) \tag{1}$$

$$\mathcal{J}_{\mathcal{L}}^{\texttt{int}}(m,t,D;\phi) = -r\log p(t'|m,t,D;\phi) + \lambda_s \log p(t'=t|m,t,D;\phi) + \lambda_{hl}H_{\mathcal{L}}(\phi) \tag{2}$$

where $H_{\mathcal{S}}$ and $H_{\mathcal{L}}$ denote entropy regularization for the speaker and listener policies respectively. $\lambda_{hs}$ and $\lambda_{hl}$ are non-negative regularization coefficients and $\lambda_s \geq 0$ is a scalar quantity.

**Behavior Grounding** If we wish to enable agents to communicate with humans, it is essential to ground their behavior into the corresponding human distribution. Recent work [25, 33, 34] achieves this by performing supervised learning with the use of a natural language dataset. Consequently, the speaker needs to send messages that are human-interpretable while the listener should be able to understand natural language. To achieve this, we let humans provide descriptions of the objects and collect such (object, description) pairs to train our agents using supervised learning. Let us denote this dataset by $\mathcal{D}$ with

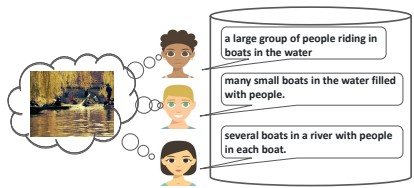

Figure 3: Limited human dataset $\mathcal{D}$.

$m^*$ being the description for the target object $t$ present in $\mathcal{D}$. Then the corresponding cross-entropy losses for the speaker ($\mathcal{J}_{\mathcal{S}}^{\texttt{sup}}$) and the listener ($\mathcal{J}_{\mathcal{L}}^{\texttt{sup}}$) are:

$$\mathcal{J}_{\mathcal{S}}^{\texttt{sup}}(m^*,t;\theta) = -\frac{1}{l}\sum_{j=1}^{l}\sum_{c=1}^{|V|}m_{j,c}^*\log p(m_{j,c}|m_{<j},t;\theta) \tag{3}$$

$$\mathcal{J}_{\mathcal{L}}^{\texttt{sup}}(m^*,t,D;\phi) = -\sum_{j=1}^{K+1}\mathbb{1}_{(t_j=t)}p(t_j|m^*,t,D;\phi) \tag{4}$$

# 3 Dynamic Population-based Meta-Learning

The previous two approaches help in learning protocols that are either closer to human prior or are ad-hoc conventions that get emerged during training. A common pitfall of training a single set of speaker/listener is that both agents tend to overfit to their partners resulting in loss of generalization across multiple partners including humans [11, 5, 29]. Since our objective is for the agents to learn robust protocols that match the human distribution and are high rewarding, we adopt a population-based approach to train agents that generalize across diverse agents in a population.

Previous approaches have used population as means to regularize the behavior of the agents using a community of fixed number of agents [48] or by meta-learning an agent on this fixed population of agents. In a given population, each agent learns a best-response to the other agent with the underlying hypothesis being the meta-agent learning the overall best-response to all agents in the population.

The main aim of meta-learning techniques [51, 45, 41, 10] is to learn a meta-agent that is able to achieve good generalization across multiple tasks. In the reinforcement learning setting, these tasks can be different sets of goals in the same or different environments. In a multi-agent interactive setting, we can reformulate these goals as generalizing to different agents in a population where the agent's partner is its environment. Consequently, meta-learning can only aim to learn generic protocols if there is a rich signal emerging from a *diverse* set of populations. A previous approach [32] (L2C) investigating learning diverse agents using different seeds for random initialization of the agent parameters. However, this simplistic way of inducing diversity is limited and only works with more toy (artificial) languages, thus leaving a big performance gap when agents communicate in natural language [33]. We hypothesize that the desired useful diversity (for meta-learning) is absent in these static populations, hence resulting in lower performance of the meta-agent.

We aim to tackle this issue by building a dynamic population in an iterative manner, similar to Policy-Space Response Oracles (PSRO) [22], which uses empirical game-theoretic analysis in a non-cooperative game to learn a meta-strategy for policy selection. An agent learns a best-response to an Oracle which is then added to a buffer that stores this growing population of agents. This buffer is used to obtain an improved Oracle which is used in subsequent iteration. We split the training into two phases as depicted in Fig 2b. We will describe the method for training a meta-listener (an equivalent method is used for the meta-speaker):

**Learning a meta-listener** We train a meta-listener that plays with all speakers present in the current population buffer. At each training iteration, the meta-listener aims to distill the behavior of all past listeners that interacted with the current speaker population.

**Building the speaker population** After obtaining the newly trained meta-listener, we expand the speaker population by playing the meta-listener with the speaker that was added to the buffer in

---

**Algorithm 1:** Algorithm

**Input** : Dataset $\mathcal{D}$, collection of objects $\mathcal{O}$, randomly initialized speaker parameters $\theta_0$, listener parameters $\phi_0$, meta-speaker parameters $\vartheta_0$, and meta-listener parameters $\varphi_0$, empty buffers $\mathcal{B}_\mathcal{S}^0$ and $\mathcal{B}_\mathcal{L}^0$

$\theta_1 \leftarrow \texttt{SupervisedLearning}(\{m,t\} \sim \mathcal{D}; \theta_0)$
  ▷ Eq (3)
$\phi_1 \leftarrow \texttt{SupervisedLearning}(\{m,t,D\} \sim \mathcal{D}; \phi_0)$
                ▷ Eq (4)
$i \leftarrow 1$
**repeat**
    $\mathcal{B}_\mathcal{S}^i \leftarrow \mathcal{B}_\mathcal{S}^{i-1} \bigcup \theta_i$
    $\mathcal{B}_\mathcal{L}^i \leftarrow \mathcal{B}_\mathcal{L}^{i-1} \bigcup \phi_i$
    $\vartheta_0' \leftarrow \vartheta_{i-1}; \varphi_0' \leftarrow \varphi_{i-1}$
    **for** $j \in \{1,2,\ldots,n_{meta}\}$ **do**
        $\vartheta_j' \leftarrow \texttt{MetaLearning}(\mathcal{O}; \vartheta_{j-1}', \{\forall \phi \in \mathcal{B}_\mathcal{L}^i\})$
                ▷ Eq (5)
        $\varphi_j' \leftarrow \texttt{MetaLearning}(\mathcal{O}; \varphi_{j-1}', \{\forall \theta \in \mathcal{B}_\mathcal{S}^i\})$
                ▷ Eq (6)
    **end**
    $\theta_0' \leftarrow \theta_i; \phi_0' \leftarrow \phi_i'; \vartheta_i \leftarrow \vartheta_j'; \varphi_i \leftarrow \varphi_j'$
    **for** $l \in \{1,2,\ldots,n_{int}\}$ **do**
        $\theta_l' \leftarrow \texttt{InteractiveLearning}(\{t,D\} \sim \mathcal{O}; \theta_{l-1}', \varphi_i)$ ▷ Eq (1)
        $\phi_l' \leftarrow \texttt{InteractiveLearning}(\{t,D\} \sim \mathcal{O}; \phi_{l-1}', \vartheta_i)$ ▷ Eq (2)
    **end**
    $\theta_0'' \leftarrow \theta_l'; \phi_0'' \leftarrow \phi_l'$
    **for** $m \in \{1,2,\ldots,n_{sup}\}$ **do**
        $\theta_m'' \leftarrow \texttt{SupervisedLearning}(\{m,t\} \sim \mathcal{D}; \theta_{m-1}'')$ ▷ Eq (3)
        $\phi_m'' \leftarrow \texttt{SupervisedLearning}(\{m,t,D\} \sim \mathcal{D}; \phi_{m-1}'')$ ▷ Eq (4)
    **end**
    $\theta_{i+1} \leftarrow \theta_m''; \phi_{i+1} \leftarrow \phi_m''$
    $i \leftarrow i+1$
**until** *performance of $\vartheta$ and $\varphi$ converge*

the previous iteration. This encourages new behavior to emerge as a result of interacting with the distilled meta-listener. Our hypothesis is that by adopting this iterative mechanism of distillation and expansion, we are able to obtain a diverse population where the behavior of each consecutive agent in the buffer changes gradually.

In this work, we use techniques that use gradient descent for optimizing the meta-agent. In particular, we use the popular Model Agnostic Meta Learning (MAML) [10] to train the meta-agents. MAML based approaches provide a good initialization over the parameters to be able to quickly adapt to new tasks. An important point to note here is that in our setup, the task distribution is changing over time and is determined as a result of multi-agent learning. We also show some results using other algorithms that are derived from MAML in the Appendix. We denote the parameters of meta-speaker $\mathcal{S}^m$ by $\vartheta$ and meta-listener $\mathcal{L}^m$ by $\varphi$. We assume a buffer of speakers denoted by $\mathcal{B}_{\mathcal{S}}$ and listeners by $\mathcal{B}_{\mathcal{L}}$. We split the collection of objects $\mathcal{O}$ into two sets $\mathcal{O}_i$ and $\mathcal{O}_o$ in order to compute the inner and outer loop losses in MAML respectively. Now, we can define the objective functions for the meta-speaker ($\mathcal{J}_{\mathcal{S}^m}^{\texttt{meta}}$) and the meta-listener ($\mathcal{J}_{\mathcal{L}^m}^{\texttt{meta}}$) as follows:

$$\mathcal{J}_{\mathcal{S}^m}^{\texttt{meta}}(\mathcal{O}; \vartheta) = \sum_{\mathcal{L} \in \mathcal{B}_{\mathcal{L}}} \mathcal{J}_{\mathcal{S}^m}^{\texttt{int}}\left(t^o; \vartheta - \alpha \nabla_\vartheta \mathcal{J}_{\mathcal{S}^m}^{\texttt{int}}\left(t^i; \vartheta\right)\right) \tag{5}$$

$$\mathcal{J}_{\mathcal{L}^m}^{\texttt{meta}}(\mathcal{O}; \varphi) = \sum_{\mathcal{S} \in \mathcal{B}_{\mathcal{S}}} \mathcal{J}_{\mathcal{L}^m}^{\texttt{int}}\left(m^o, t^o, D^o; \varphi - \alpha \nabla_\varphi \mathcal{J}_{\mathcal{L}^m}^{\texttt{int}}\left(m^i, t^i, D^i; \varphi\right)\right) \tag{6}$$

where $t^i \in \mathcal{O}_i$, $t^o \in \mathcal{O}_o$, $m^o = \mathcal{S}(t^o)$, $m^i = \mathcal{S}(t^i)$, and $D^i$ and $D^o$ are sets of distractor objects sampled from $\mathcal{O}_i$ and $\mathcal{O}_o$ respectively.

Finally, the trained meta-agents are fine-tuned using the dataset $\mathcal{D}$ before the testing phase. The fine-tuning losses are the same as the supervised losses Eq (3) (4) described in the previous section. Since the number of training iterations is unknown and could be potentially much larger than the size of the buffer the memory can hold, we use reservoir sampling to keep a uniform sample of past agents in the buffer. The detailed algorithm can be found in Algorithm 1.

## 3.1 Prior approaches

Recent work has investigated combining the two objective functions of self-play and imitation learning, with the goal of training agents that use natural language and perform well on a given emergent communication task. This transforms the problem into training a task conditional language model in a multi-agent setup. S2P[33, 13] proposes methods that devise a curriculum between the two training phases updating the speaker and the listener in an iterative manner. SIL[34] follows a student-teacher paradigm that is trained sequentially, where the teacher agents, initialized from the student agents, are trained using interactive learning. Then the student agents are trained to imitate the teacher agents by sampling data at every training iteration. Another work by [25] investigates different types of language (semantic and structural) and pragmatic drifts. They propose a reranking module that first samples multiple messages from the speaker and then ranks them according to the task performance.[2] Crucially, the reranking module presents an orthogonal axis of progress and thus can be combined with other approaches presented here.

With respect to training agents using population-based methods, [48] propose a learning method that uses a community of fixed number of agents where speakers and listeners are randomly paired with each other at every training iteration. L2C[32] proposes a meta-training method on a fixed population of agents. They first train different populations of agents via self-play with each population initialized with a different random seed. Then a meta-learner interacts with these agents and learns to adapt to each population simultaneously. Another similar method in [5] aims at learning a community of agents, where groups of speakers and listeners are used to sample a pair uniformly at random who then play the game and jointly optimize for better task performance. During learning, few agents are reinitialized periodically/at random from a group of agents. The idea is to promote cultural transmission to induce compositionality over multiple generations of language transfer. For our experiments, we reinitialize agents to the pretrained agents that are trained using the human dataset. We denote this method as GEN.TRANS. in the following sections.

---

[2]Their approach uses a different setup that involves giving speaker access to the distractor objects.

# 4 Experiments and Results

In this work, we conduct experiments on two referential games in two different modalities, i.e., having agents communicating in English about images, a process akin to image captioning, and having agents communicating in German about English sentences, a process akin to machine translation.

**Images**   We use the image-based referential game [27, 23, 33, 25] which is a common environment used to analyze emergent protocols involving multimodal inputs. We use the MSCOCO dataset [30] to obtain real images and the corresponding ground truth English captions annotated by humans. Since this dataset has multiple gold captions for each image, we randomly select one from the available set and keep $|\mathcal{O}| = 7000$. Following [26, 33], both speaker and listener are parameterized with recurrent neural networks (GRU [4]) of size $512$ with an embedding layer of size $256$. We embed images using a Resnet-50 model [16] (pretrained on ImageNet [7]). We set the vocabulary size to $100$ and the maximum length of the sentences at $15$. We use the same speaker and listener buffer size of $200$ for reservoir sampling. Other implementation details are given in the Appendix.

**Text**   To further test the robustness of our approach to different types of structural priors found on the input data [46], we also play a referential game using text as the input modality [26]. We use the publicly available IWSLT'14 English-German dataset with English as the source and German as the target language. Instead of randomly selecting the distractors from the set of English sentences, we create harder distractors by picking English sentences that are more similar to the source English sentence. For this, we use a Sentence-BERT model [43] (pretrained on SNLI [2] and MultiNLI [52]) to embed all English sentences in the dataset and filter them using a threshold criteria of cosine similarity to systematically choose the distractors. The game complexity increases with an increase in cosine similarity, and we fix this to $0.85$. We use a vocabulary size of $100$ and maximum sentence length of $20$. All other details are the same as the game with images.

Throughout this section, we report results using 3 random seeds and $K = 9$, $|\mathcal{D}| = 5000$ in the image game and $K = 14$, $|\mathcal{D}| = 5000$ in the text game. In the Appendix, we show that our findings are robust and include results with $|\mathcal{D}| = 2000$ and $K = 9$ on the image and text game respectively.

## 4.1 Referential Accuracy

We report referential accuracy as the final task performance of the meta-agents, i.e., how accurately the (meta-) listener is able to predict the target using the (meta-) speaker's messages. Fig 4 plots the referential accuracy on the test set of 1000 images/sentences for the image/text game. In both plots, the pretrained baseline is obtained by separately pretraining the speaker and the listener on the language data without any interactive or meta-learning signal, i.e., only using Eq 3 and Eq 4, and then pairing them together. The emergent communication (EMECOM) baseline is computed by training agents purely via interactive learning and without any supervision from the dataset, i.e., only using Eq 1 and Eq 2. In both image (left plot, $10\%$ random baseline) and text (right plot, $6.7\%$ random baseline), our method outperforms alternative approaches. Specifically, GEN.TRANS.

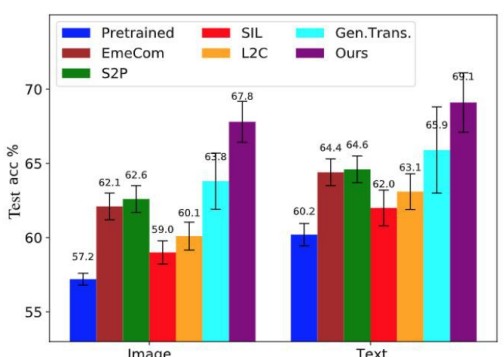

Figure 4: Referential accuracy of the (meta-) agents on the test set (Left): Image game and (Right) Text game ($p < 0.05$, t-test).

and our method outperform S2P and SIL, both of which can be thought of as single-agent population, indicating the importance of diversity among the agent population. Moreover, our method surpassing GEN.TRANS. and L2C indicates the importance of using the proposed meta-learning approach in conjunction with an adaptive population instead of using a static set of agents. In Sec 4.4 we conduct more ablations to better assess the importance of the different components for our approach.

## 4.2 Evaluating Natural Language Skills of (Meta-) speakers

Previous work on multi-agent communication with language has showed that even when agents receive supervision using natural language data, language drift phenomena result in agents noticeably diverging from natural language while still achieving high referential accuracy [25, 26]. As such, we proceed with directly evaluating the natural language capabilities of the meta-speakers. We start with the image game, in which we view the utterances generated by the meta-speaker as captions and hence evaluate their alignment of the generated captions with human generated ones. We evaluate all methods using 1000 generated captions from the test set. We report BLEU score [39] between the generated captions and the ground-truth English caption. We also report an alternative caption score proposed by [6], which uses a pre-trained model on MSCOCO to compute a similarity score between the generated caption and the context (image and ground-truth caption). The results are shown in Table 1, where we see

|  | Caption Score | BLEU |
|---|---|---|
| PRETRAINED | $5.2 \pm 0.02$ | $24.2 \pm 0.1$ |
| S2P | $5.5 \pm 0.04$ | $26.1 \pm 0.3$ |
| SIL | $5.5 \pm 0.04$ | $25.6 \pm 0.24$ |
| L2C | $5.4 \pm 0.07$ | $25.4 \pm 0.45$ |
| GEN.TRANS. | $6.1 \pm 0.05$ | $28.4 \pm 0.32$ |
| **OURS** | $\mathbf{6.6} \pm 0.04$ | $\mathbf{29.3} \pm 0.25$ |

Table 1: Evaluating the (meta-) speaker in the image game using the metric proposed in [6] ($p < 0.05$, t-test).

that our method outperforms alternatives indicating that the use of meta-agents is effective in getting higher referential accuracy but not on the expense of worse language skills.

In Table 2, we perform a similar analysis for the (meta-) speaker on the text game. Here, the utterances of the meta-speaker could be thought of as translating the English input sentences to German. Hence, we evaluate the (meta-) speaker as an English-German machine translation system computing BLEU score with the German reference sentences. Apart from comparing our model against different alternatives, we also create stronger baselines by using the English-German pretrained model in the text game with different decoding strategies (all others methods including ours will use greedy decoding). We find that our method, despite using greedy decoding, is able to outperform all variables of the PRETRAINED model, indicating that our technique has a somewhat different effect than the one introduced by simply changing the decoding strategy.

|  | BLEU |
|---|---|
| S2P | $21.7 \pm 0.22$ |
| SIL | $21.6 \pm 0.21$ |
| L2C | $21.1 \pm 0.29$ |
| GEN.TRANS. | $23.9 \pm 0.21$ |
| PRETRAINED | |
| + Greedy Decoding | $20.8 \pm 0.06$ |
| + Beam Search ($n = 2$) | $21.1 \pm 0.06$ |
| + Beam Search ($n = 4$) | $22.4 \pm 0.05$ |
| + Top-k Sampling ($k = 40$) | $24.6 \pm 0.05$ |
| **OURS** (using greedy decoding) | $\mathbf{25.2} \pm 0.16$ |

Table 2: BLEU score on German sentences for (meta-) speaker in the text game across various baselines and decoding strategies. $n$ denotes number of beams in beam-search ($p < 0.05$, t-test).

All in all, we find that even though our objective task was never to just simply maximize captioning (or machine translation) performance, our dynamic population-based meta-learning resulted in meta-speakers with better language skills than the pretrained models, which use the very same number of (limited) language data. While our results are far from being state-of-the-art on captioning or translation, we nevertheless see that our technique can be thought of as a form of semi-supervised learning through self-play, and can perhaps be used in combination with specialized models of the respective tasks. We also include some qualitative samples generated by different speaker models alongside the ground truth captions in Fig 8. In the Appendix, we show some additional corpus-level statistics that further corroborates this claim.

## 4.3 Agents interacting with Humans

Although BLEU score is able to capture some form of syntactical and semantic drift, it still fails to counter the phenomenon of pragmatic drift, as introduced in [25]. For this reason, we evaluate the performances of our agents, and all other alternative methods, by having agents interact with humans and report referential accuracy. As such, we play games where the (meta-) speaker interacts with a human listener and the (meta-) listener with a human speaker. Thus, in the first case we evaluate the natural language generation abilities of the (meta-) speaker, whereas in the second

case we evaluate the natural language understanding skills of the (meta-) listener. The (meta-) speaker is evaluated using 1000 games with humans and the (meta-) listener using 400 games.[3]

Moreover, we compute an upper-bound by computing the referential accuracy when pairing two humans to play the game. In Fig 5a, we show the results for meta-speaker in the image game where our method outperforms other approaches by a significant margin. In Fig 5b we compare the performance of the meta-listener in the image game with other baselines and found similar observations to the meta-speaker. We find that our method is able to

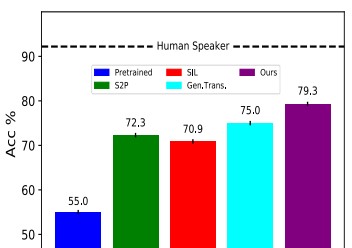 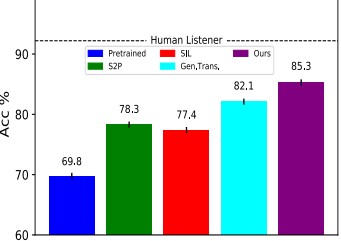

(a) Agent speaker vs Human listener  (b) Agent listener vs Human speaker

Figure 5: Human evaluation on the image game. The black line in both plots represent the performance of the (Human-speaker, Human-listener) pair.

better understand human descriptions by learning diverse caption representations. The results for the game with text follow similar pattern and can be found in Appendix along with other details of the experimental setup. Overall we infer that our method suffers the least amount of pragmatic drift as compared to other baselines, as measured by the performance gap with the human-human gameplay.

## 4.4 Ablation studies

We further analyze the importance of each component of our proposed algorithm by introducing the following ablations.

**no meta-agents** In this ablation, we test the importance of building populations of agents by learning speakers (or listeners) through interaction *with the meta-agent* listener (or speaker respectively), as opposed to just interacting with a randomly chosen listener (or speaker) from the buffer. Our hypothesis is that this promotes diversity within the population, resulting in learning more general communication protocols. Before commenting on the performance of this ablation, we note that this type of training is in fact more unstable, which we attribute to making the training of the two meta-agents interdependent of each other. We leave further analysis of this for future work.

**no adaptive-meta** In this ablation, we test the importance of *training the meta-agent in an iterated manner*, as opposed to training it from scratch by resetting the weights of the meta-agent before interacting with the population. We build the population of agents that interact with the meta-agent (whose weights have been reset) in two different ways: (i) by following the approach illustrated in Alg 1 and (ii) by following the approach introduced in No meta-agents above.

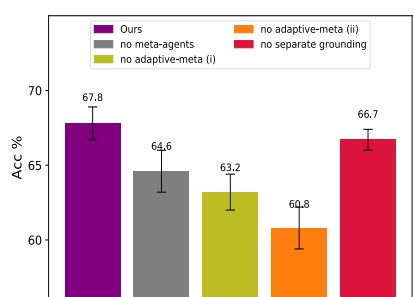

Figure 6: Referential accuracy for different ablations in the image game.

**no separate grounding** In this ablation, we test the importance of avoiding catastrophic forgetting of language skills by conducting interactive and supervised learning (i.e., Equations 1 till 4 in Algorithm 1) in separate phases, in line with previous work [33]. Recently, other works [25, 34] have proposed using KL divergence between the agents' policies with a pre-trained policy on

---

[3]We note that the participants were neither aware of the identity of the agent they were playing against nor they played consecutive games with the same agent. This allows for a more fair comparison of the language skills of agents since participants did not have the chance to adapt to ad-hoc conceptions potentially used by agents (e.g., consistently referring to cats as onions).

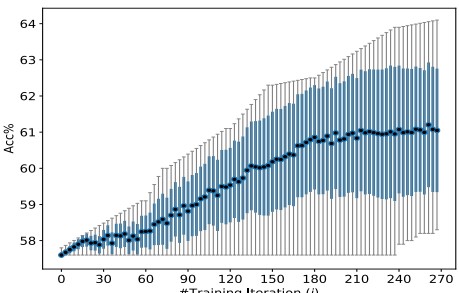

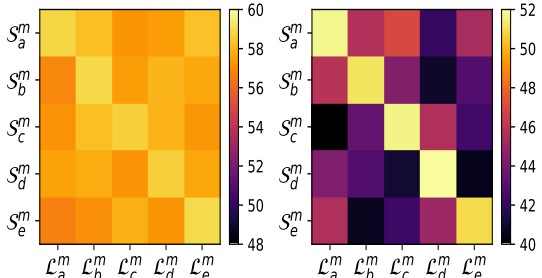

Figure 7a: Average referential accuracy on the test set when a trained meta-listener individually plays with each speaker present in the buffer at the corresponding training iteration. The blue bars show the standard deviation across all speakers in the buffer. The gray bars show the interval between the minimum and maximum performing speaker.

Figure 7b: Cross-play evaluation using referential accuracy for our method (Left) that dynamically builds the population and L2C (Right) that uses static population, across 5 sets of meta-speakers and meta-listeners. Left: Our meta-agents have much less variance across various partners. Right: L2C meta-agents suffer from high variance and overfitting to their own partners (diagonal). The scale of the colorbar is calibrated based on the highest variance interval in both plots to allow fair comparison between the two methods.

language data, and as such penanalize policies that diverge from language. Concretely, the corresponding speaker objective is $\lambda_{int}\mathcal{J}_{\mathcal{S}}^{\texttt{int}} + KL(\theta_{\text{Pretrained}}||\theta)$.

In Fig 6, we compare the performance of our full approach against all ablations by reporting the referential accuracy on the test set in the image game. The comparatively lower performance of no meta-agents against our method indicates that the agents that interact with the corresponding meta-agent learn more robust and diverse strategies as compared to the ones that interact with any agent in the buffer. Both variants of the no adaptive-meta agents underperform against our approach as well as against no-meta agents. This suggests that the *adaptive* way of training the meta-agent is crucial in getting higher performance as compared to training it from scratch. We hypothesize that our iterated meta-agent has a slight advantage of being able to learn from agents that get created during the training in turn providing a better initialization to adapt to the new population. The meta-agent captures useful information from past iterations of other agents and provides a richer learning signal to train the next agent to be added. No separate grounding agents perform close to the alternating updates used in our method indicating that one can use multiple ways to integrate the two loss functions in combination with our meta-learning approach.[4]

## 4.5 Induced diversity and out-of-population evaluation

In Fig 7a, we show the average performance of the meta-listener, obtained after training, playing with different speakers at different stages of their training in the image game. We show that as the training of the speakers progresses, the standard deviation (denoted by blue bars) of the referential accuracy across all the speakers present in the buffer up to that training iteration increases, together with the difference between the best and worst performing agents (denoted by gray bars). This indicates that as the population grows the diversity among the agents also improves, thus resulting in richer training signal for the meta-agents. In the Appendix, we include a similar plot for the meta-speaker reporting average BLEU score.

We also perform cross-play evaluation to test the generalization during out-of-population communication. For doing so, we obtain 5 meta-agents (5 meta-speakers and 5 meta-listeners) by using Algo 1 with 5 different seeds and report referential accuracy on the test set when pairing all possible combinations of meta-speakers and meta-listeners. On the left plot of Fig 7b we observe that, as expected, the in-population communication found on the diagonal is the best, and performance barely degrades when pairing agents that were not initialized with the same seed, indicating that our agents have learn more generalizable conventions. On the other hand, when conducting a similar evaluation with L2C (right plot), we observe that the protocols learnt in this case overfit to their own partners

---

[4]Note that our choice of using alternating updates for interactive learning and supervised learning is arbitrary and based on [33]. We believe that any multi-objective learning method could work here.

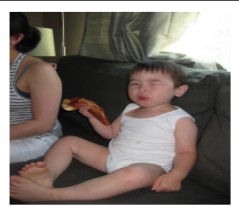
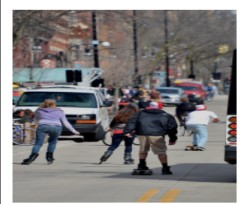

**Human: A boy in a white dress sitting on a black couch while holding a pizza.**

S2P: A boy in a white shirt eating pizza while sitting.
SIL: A boy in a white dress sitting on a bed with pizza.
L2C: A boy in a white dress eating pizza and pizza.
Gen.Trans.: A boy in a white dress sitting bed and bed and bed...
*Ours: A boy in a white dress holding a pizza sitting on a black couch.*

**Human: People on roller-skates riding on the road with a white car in the background.**

S2P: A group of people with roller-skates on the road.
SIL: People running with roller-skates on the road.
L2C: A group of people walking on the road.
Gen.Trans.: A group of people with roller skates road car road car...
*Ours: A group of people with roller-skates on the road and a white car.*

Figure 8: Qualitative samples generated by the (meta-) speaker in the game with images.

(diagonal) resulting in high variance across other players, and thus indicating the adoption of more ad-hoc protocols. Consequently, we posit that our method be used for few-shot generalization to unseen partners in a multi-agent interactive environments.

### 4.6 Robustness against implicit bias in the dataset

As agents trained using a single dataset may overfit to the objects present in that particular dataset, in this section we design an experiment to test the agents' robustness against implicit dataset biases. We conduct a *cross-task* evaluation experiment on the referential game with $K = 9$ distractors where we compute the referential accuracy using images from the Flickr8k dataset [17] while the agents were trained using the images present in the MSCOCO dataset. Similarly, we also perform a *within-task* evaluation where we train and then evaluate our agents using the same Flickr8k dataset. We use 5000 images as training set and 1000 images as validation/test set.

Results are presented in Table 3. While we do observe a drop in the performance when compared to the within-task performance (70.2 v/s 78.9), our across-task model still outperforms the within-task PRETRAINED baseline (70.2 v/s 65.3). This observation correlates with our results using human players and out-of-population evaluation shown in Sec 4.3 and Sec 4.5 respectively.

|            | PRETRAINED     | OURS           |
|------------|----------------|----------------|
| Cross-task | $59.7 \pm 0.9$ | $70.2 \pm 1.8$ |
| Within-task| $65.3 \pm 0.3$ | $78.9 \pm 1.4$ |

Table 3: Referential accuracy on the test set showing robustness against implicit bias in the dataset.

## 5 Conclusion

We presented a dynamic population-based method to train agents in a cooperative multi-agent reinforcement learning setup. We showed that our method induces useful diversity into a population of agents which helps in learning a robust meta-agent. Empirically, we show that our agents outperform prior work on the task performance, auxiliary tasks and even human-based evaluation. In the future, we plan to extend this approach in cases where agents are performing temporally extended physical actions in an environment with perceptual observations. One limitation of our approach pertains to maintaining a buffer of past agents that are used to train the meta-agent. Moreover, we would also like to analyze the effect of using task-independent language data (as opposed to task-dependent that we use now) within a multi-agent reinforcement learning task.

## Funding Transparency Statement

AG is supported by IBM scholarship. We thank Compute Canada for providing the compute resources to run the experiments.

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
