

Figure 9: Detailed algorithm outline. We show the different phases involved in training a meta-listener $\mathcal{L}^m$ and building the corresponding speaker population (buffer) $\mathcal{B}_\mathcal{S}$. Similar outline can be drawn for training the meta-speaker $\mathcal{S}^m$.

## A   Hyperparameters and other training details

We show here the range of parameter configurations we tried during training (bold indicates the ones used in the experiments):

We use the Adam optimizer [20] in PyTorch [40] for training the agents. For the baselines (S2P, SIL, L2C, GEN.TRANS.), we used the publicly available repositories attached with the respective publications. We adapt their codebase to train agents on the two referential games used in this work while tuning the hyperparameters separately for each method and each game keeping the same architecture across all baselines. We even used a larger batch size for some methods that performed better than the ones reported in the original papers. We chose the hyperparameters by performing a grid search over the values mentioned in Table 4 and others in §4.

| Name | Values used |
|---|---|
| Batch Size | 512, **1024** |
| Buffer Size | 50, 100, **200** |
| $n_{meta}$ | 20, 40, **60**, 65, 70 |
| $n_{sup}$ | 10, 20, **25**, 30 |
| $n_{int}$ | 40, 60, 70, **80**, 100 |
| $\lambda_{hs}$ | 0.1, **0.01**, 0.001 |
| $\lambda_{hl}$ | 0.1, **0.03**, 0.007, 0.001 |
| $\lambda_s$ | 0.1, 0.5, **0.8**, 1 |
| Learning rate (outer loop) | **1e-4**, 1e-5, 6e-5, 6e-4 |
| Learning rate (inner loop) | **1e-4**, 3e-4 |
| $\lambda_{int}$ | 1, **0.1**, 5 |

Table 4: Hyperparameters. Bold indicates the chosen values used for the final analysis.

We used the pretrained Resnet-50 embeddings of dimension 2048 for the image game and Sentence-BERT embeddings of size 768 for the text game. We used our internal cluster consisting of Nvidia V100 GPUs to train the models. The annotations in the MSCOCO dataset belong to the COCO Consortium and are licensed under a Creative Commons Attribution 4.0 License.

# B Further Results

In this section, we show results on the text game, ablation using meta-learning methods, and additional metrics to evaluate the natural language skills of the (meta-) speakers.

**Human Evaluation** We used 9 distractor objects and the models trained with $|\mathcal{D}| = 5000$ for both games. For the image game, in Fig 5a each agent speaker outputs 1000 utterances corresponding to all the images in the test set, which are then given to the human listeners to play the game. Similarly, for evaluating the agent listener with a human speaker, each agent evaluates 400 human utterances in Fig 5b. The black line corresponds to 400 human-speaker vs human-listener matches. In Fig 10,

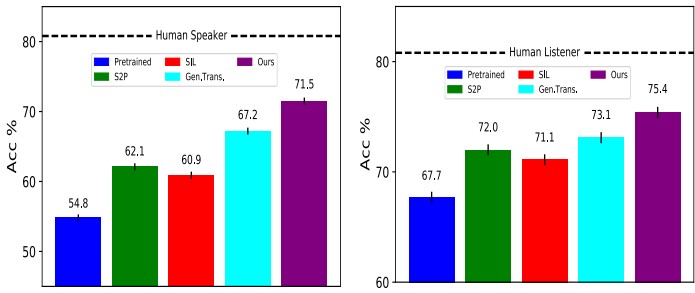

(a) Agent speaker vs Human listener (b) Agent listener vs Human speaker

Figure 10: Human evaluation on the text game. The black bar in both plots represent the performance of the (Human-speaker, Human-listener) pair.

we present the results of the human evaluation on the text game. Similar to the trend found in Sec 4.3, we show that agents trained using our method beat all prior baselines when paired with both human listeners and human speakers. Both Fig 10a and 10b are drawn using 100 agent (and human) utterances that translate an English sentence to German.

**Referential Accuracy** We show further results for both games in addition to the results found in § 4.1. Fig 11a shows results for the configuration $|\mathcal{D}| = 2000$ and $K = 9$ in the image game. EMECOM results in the highest performance here. We argue that even though its performance on referential accuracy is higher than our method, the corresponding BLEU score (or its compatibility with a human partner) is $\sim 0$.

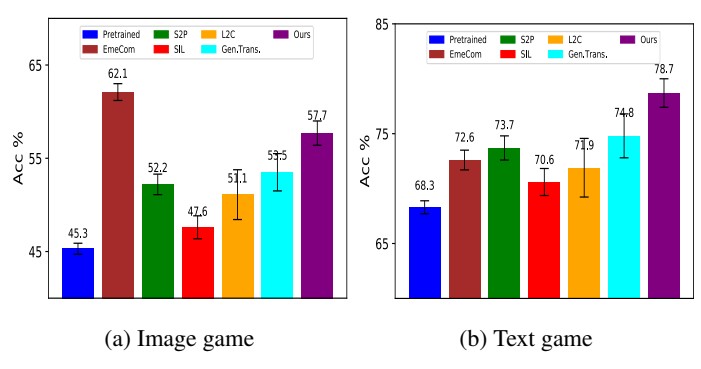

(a) Image game      (b) Text game

Figure 11

This means that it does not generalize to out-of-population agents or human partners and rather learns ad-hoc conventions that only generalize to its own partner, in line with previous work in emergent communication [25]. Moreover, this behavior being only observed in $|\mathcal{D}| = 2000$ case but not in the $|\mathcal{D}| = 5000$ case indicates the existence of a threshold (in the # human samples) above which the EMECOM baseline would underperform against other methods, given the same task/network configuration. Fig 11b shows results on the text game with $|\mathcal{D}| = 5000$ and $K = 9$. Similar to the trend observed in Fig 4, our method outperforms all other baselines.

**Induced Diversity** In Fig 12a, we plot the average BLEU score of a trained meta-speaker, obtained after training, playing with different listeners at different stages of the training in the image game. The blue bars show the standard deviation across all agents present in the buffer. Similar to the observations in §4.5, we find that the natural language skills of the meta-speaker improve as the training progress while the population still being diverse enough to provide rich learning signal for meta-training.

**Other meta-learning methods** We also performed an ablation study using different meta-learning algorithms [10, 38]. FOMAML is the first-order approximation of MAML and Reptile is another first-order meta-learning algorithm that performs stochastic gradient descent for a few steps across all tasks and then updates the parameter towards the average of updated task-specific weights. We show the results in the image game using referential accuracy in Fig 12b. The performances of the all algorithms are competitive with each other indicating robustness across the three methods. Furthermore, we think that recent advancements in meta-learning algorithms [44, 36] could be combined with our algorithm to further analyze this effect and investigate biases resulting from a specific meta-algorithm.

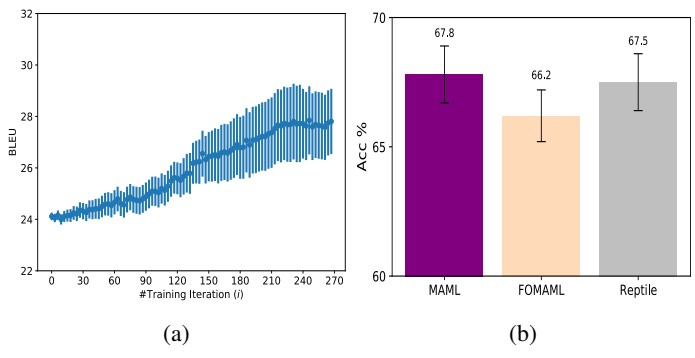

(a)     (b)

Figure 12

**Natural language skills of meta-speakers** We also show some corpus-level statistics for studying the linguistic diversity of the produced messages over and above the BLEU statistics shown in Sec 4.2.

- Average ratio of the length of generated utterance per human utterance We analyze the average sentence length for all the generated utterances in the test set and compare that with the ground-truth utterances present in the dataset for the image game.

| | |
|---|---|
| S2P | $0.62 \pm 0.05$ |
| SIL | $0.63 \pm 0.05$ |
| GEN.TRANS. | $0.58 \pm 0.04$ |
| **OURS** | $0.68 \pm 0.05$ |

- Average ratio of unique words in generated utterance per human utterance We also study the number of unique words in an utterance in the image game.

| | |
|---|---|
| S2P | $0.67 \pm 0.02$ |
| SIL | $0.69 \pm 0.02$ |
| GEN.TRANS. | $0.7 \pm 0.03$ |
| **OURS** | $0.8 \pm 0.03$ |

We would like to point out that the utterances in the dataset were collected for a different task (image captioning and machine translation). Hence the generated captions are less diverse and shorter as compared to the human captions because the underlying interactive learning task only requires *captioning in context*.

## C    Human Experiment Setup

Our human experiments were done in a controlled environment with a group of 45 undergraduate and graduate students. The experiments were overseen and approved by our internal lab review board. The participants were not compensated for taking part in the experiments as our lab has been conducting such experiments in the past on a quid pro quo basis. Moreover, the participants were given the following instructions to play the game and were ensured that their individual identities would not be revealed or used in a way that could contaminate our results. Consequently, there were no participant risks involved in our experiments. In addition, to filter noise in the experiments, for each utterance we evaluated the performance of each participant against other participants. Concretely, we played each utterance of the (speaker) participant with 5 other (listener) participants and compared the performance across all 5 games. All utterances that do not obtain the same game score for at least $4/5$ games were excluded. Further filtering was done based on a threshold given by the BLEU score (threshold=30) between the participant utterance and the ground-truth utterance.

Overall instructions:

*We are interested in conducting human experiments for the popular referential games proposed in Lewis et al, '69. It is a cooperative game that involves two players: a speaker and a listener. A speaker observes a target image and emits a message that is sent to the listener. The listener looks at the message and tries to identify the target image among a set of distractor images. Both agents are given a positive reward if the listener's prediction is accurate and zero otherwise. Our experiments*

*require each participant to play as a speaker and a listener with different partners. Your partner could be a human or one of our trained agents. You will not be given the identities of your partners and your individual identities will not be used for analyzing the final results.*

Message to the human speaker:
*You are assigned the role of a speaker. Look at the image carefully and write a caption that best describes the image.*

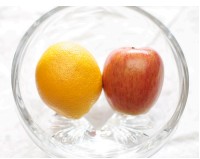

Message to the human listener:
*You are assigned the role of a listener. Read the message carefully and use that to choose the target image for which the message was intended, among the set of other distractor images.*

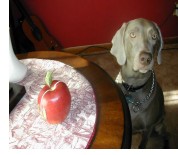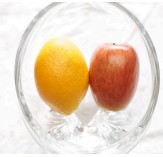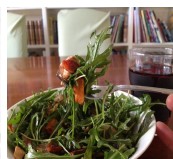

## D  Failure examples

We show some qualitative failure examples from the image game where our agents exhibit different kinds of errors. We group all of them into three categories:

- Incorrect message due to imperfect vision
  Dataset utterance: *a big black bear that is walking into the road*
  Speaker message: *A black dog crossing the road with cars.*
  In this test example, the speaker incorrectly uses dog for a bear.

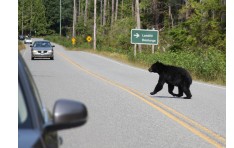

- Incorrect language usage by the speaker
  Dataset utterance: *a young man with a lacrosse stick and nike bag dressed in a shirt and tie.*
  Speaker message: *A group of people standing with bags, tie, and bottle.* Here, the speaker tried to add all observed objects in the message without being semantically correct.

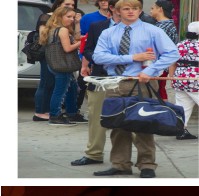

- Incorrect language understanding by the listener
  Speaker message: *A child with a teddy bear.*
  Listener chooses an incorrect target image that only contains teddy bears confusing the teddy bear with a child.

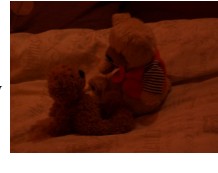

We believe that the speaker gets confused when it encounters an out-of-vocabulary object in the target image. Likewise, the listener chooses a wrong target image when the complexity of the distractors is increased (by using distractors with similar objects as in the target image). Nonetheless, we think these issues are not specific to our agent and would arise in any interactive learning model as we scale up to include more and more objects.