# OpenReview forum: "Dynamic population-based meta-learning for multi-agent communication with natural language"
_NeurIPS.cc/2021/Conference — NeurIPS 2021 Poster_

### Official Review · Reviewer_rond · 2021-07-14

**Rating:** 9
**Confidence:** 1

**Summary:**

The paper is motivated by group coordination through natural language and creates a goal of developing agents that can communicate effectively with other agents and humans. They build off of work on emergent communication and their method is most related to L2C [Lowe19], which uses a meta-learning approach to learn communication protocols.

Their training method is as follows:
```
Train with some human data
while not_converged:
    for n_meta steps:
        Learn a meta-listener on a population of speakers
    for n_selfplay steps:
        Learn new speakers through self-play with the meta-listener
    for n_supervised:
        Fine-tune on human data
train
```

And their evaluations are:
1. Image based referential games
2. A text based referential game that they design where agents are trained to communicate in German about English sentences

Their experiments show that:
1. They outperform baselines on referential accuracy in both modalities
2. They have a high sentence quality (BLEU, caption score) wrt baselines
3. The meta-speaker in the text task works reasonably effectively as an English to German translation system (BLEU)
4. Their speakers and listeners perform better on human evaluation in the image task

Learning to learn to communicate. Lowe et. al., 2019.

**Ethical Concerns:**

I do not think their work presents ethical concerns at this time.

**Limitations And Societal Impact:**

I could not find where the limitations were addressed. It would be helpful to see a discussion that describes in more detail the benefits of their framing (emergent communication instead of something drier (see main review).

**Main Review:**

Originality: The paper directly extends past work. The novelty in their training procedure comes from repurposing meta-speakers and -listeners for population generation. Their experiment design is creative: they introduce a new task that serves allows them to evaluate speakers in a novel way (as translation engines).

Quality: The thoroughness and rigor of the experiments are high. They include p-values, extensive ablations, and reasonable comparison to past work.

Clarity: The literature was well presented. By integrating related work into introduction/background on past methods (sections) their method was easier to understand. Some training details were less straightforward. E.g., in the prose it could state more explicitly how the meta-speaker and meta-listener are used together and to my knowledge the initialization is only mentioned in the algorithm box.

Significance: This paper brings studies on emergent communication in line with tasks of interest (communication with humans because of the evaluation, nlp tasks because of the translation system evaluation). One open ended question that the authors may be interested in addressing (perhaps verbally): what is the advantage of conceptualizing these training methods as emergent communication among a population of agents? E.g., a different conclusion from this paper is that using meta-learning as a distillation method across re-initializations is an effective method for solving various language tasks.

Minor comments to authors:
- L15 your -> our?
- L26 like these agents -> these agents
- L52 the use -> the use of
- It would be helpful to label Figure 1 with the math in the prose
- L162: Out -> Our

**Time Spent Reviewing:**

5

---

> ### Author Response · Authors · 2021-08-10
> **Response (1/1)**
>
> We appreciate the positive feedback of the reviewer and their insightful suggestions. We address the questions as follows:
>
> what is the advantage of conceptualizing these training methods as emergent communication among a population of agents?
> >Indeed, we agree that our proposed approach is generic and can be applied to any task that requires the agent to exhibit human-like behavior while solving a given downstream task with access to a set of human demonstrations. We believe the main contribution comes from iteratively building the population which induces constructive diversity that is further exploited by the meta-agent. In that respect, our method can also be used to distill across a widely distributed set of initializations by building an iterative population instead of using the whole population at once. In this work, we chose a communication environment as our test environment as it is an important and widely accepted testbed for learning tabula rasa agents with linguistic properties. In addition, the question of learning agents that speak natural language and are capable of performing a given interactive task has gained a recent surge and we show that our method advances improvements over prior work in this domain. Nonetheless, we agree that any cooperation task could have been used here.
>
> $~$
> how the meta-speaker and meta-listener are used together
> >During training, the meta-speaker and meta-listener do not interact with each other and are trained independently. In Alg 1, we show pseudo-code for training both meta-agents while the rest of the main text and Fig 2b describe learning a meta-listener as noted in L152-154. The role of the meta-agent is to summarize the past behavior of the agents in the buffer as also described in the above response to R3.
> During testing, we play matches with a trained meta-speaker and a meta-listener to compute the referential accuracy.

---

### Official Review · Reviewer_V9jU · 2021-07-15

**Rating:** 6
**Confidence:** 3

**Summary:**

This work deals with the problem of multi-agent communication where the agents can be artificial or human. The latter means that the conversation has to be in natural language. New agents can be added or removed from conversation so the environment is not static. The authors propose a dynamic population-based meta-learning approach. They analyse the language skills of agents using BLEU, the quality of interaction in a human trial, as well as cross-play with unseen partners achieving overall good results.

**Ethical Concerns:**

I don't see any ethical concernes.

**Limitations And Societal Impact:**

This is difficult to judge as it is not elaborated what this technology can be used for. The test cases presented do not pose any negative social impact as far as I can see.

**Main Review:**

It would be nice to give some more context to this work. Where do authors see this technology to be used? The test-beds are clearly toy examples and not very relevant. Do they see the use of these agents as bots that engage in eg Twitter communication? Or is this meant to improve two-party dialogue?

When we talk about communication and dialogue what we have in mind is multi-turn communication. The test-beds considered here focus only require singe turns so it’s not quite clear to me how much of the communication management behaviour is learned or can be learned. Note the vocabulary of the test-beds are very small, only 100 words. Can this approach scale?

The issue of an unseen and seen interlocutor is incredibly important and I would be keen to see how this method generalises to (or improves) RL approaches to task-oriented dialogue. Ablation studies show that removing meta-agents does not cause a drastic decrease in performance and I wonder if the reason is the simplicity of the task or limited use of this method. In any case it would be important to investigate this.

A lot of details are in the Appendix that would be better suited in the main text, in particular the qualitative samples.

----

The authors addressed my concerns to some extent so I'm willing to increase my score to 6.

**Time Spent Reviewing:**

2

---

> ### Author Response · Authors · 2021-08-10
> **Response (1/1)**
>
> We thank the reviewer for their helpful comments and address the concerns as follows:
>
> Technology usage, connections to dialogue works, and scalability
> >The referential game used in this work is a variant of the Lewis signaling game [26], which has been extensively used in linguistic and cognitive studies in the context of language evolution [Briscoe, 2002; Cangelosi et al., 2002; Steels et al., 2012; Spike et al., 2016] and also serves as a minimal test-bed of multi-agent and human-agent communication [1, 11, 14, 21, 22], for which we show that our work performs better than previous work on this topic.
> There is a recent surge in AI chatbots being deployed in support roles for question-answering (i.e. tech support) so it is important that their responses are natural and that they understand human motivations. Some prior work has tried to train such conversational agents and a major chunk of them use self-play which utilizes the goal-oriented reward [Lewis et al., 2016;, Kang et al., 2019; Roller et al., 2020]. Although these approaches perform well on the given dialogue task, they only use self-play with a single and static population of agents. In this work, we propose a dynamic population-based method that improves the performance of self-play while also achieving better sample efficiency than the dialogue models.
> Even though our work uses single-turn communication, we believe that it is a building block to training agents capable of conversing in multiple turns. Furthermore, despite the simplicity of the task, current agents are not capable of communicating with humans or unseen interlocutors.
> Our method does not make any hard assumptions on the vocabulary size used in the experiments. Hence there is no reason to believe that our method will not scale to a larger vocabulary. Using a smaller vocabulary set does make the RL optimization problem easier but recent work has proposed techniques such as using a reranking module [23] which allows us to use a larger vocabulary of size as large as 2000. A benefit of our modular learning algorithm is that it allows us to replace the speaker learning module with any other desired module without affecting the whole learning process.
>
> - Cangelosi et al. 2002. Simulating the evolution of language.
> - Briscoe 2002. Linguistic evolution through language acquisition.
> - Steels et al. 2002. The grounded naming game.
> - Spike et al. 2016. Minimal requirements for the emergence of learned signaling.
> - Kang et al. 2019. Recommendation as a Communication Game: Self-Supervised Bot-Play for Goal-oriented Dialogue.
> - Roller et al. 2020. Open-Domain Conversational Agents: Current Progress, Open Problems, and Future Directions.
> - Lewis et al. 2016. Deal or No Deal? End-to-End Learning for Negotiation Dialogues
>
> $~$
> Importance of meta-agents
> >We note that in Fig 6, there is a $>3$% performance gap between the “no meta-agents” ablation and our method which is significant as compared to other baselines. We did not expect a drastic decrease in the performance since a major part of our approach is the iterative building of the population also tested in another ablation “no adaptive-meta”. In the no meta-agents case, instead of the meta-listener we randomly choose a listener from the listener buffer to build the population. Now in the absence of a distilled meta-agent, we would need some other source for inducing diversity in the population. We believe that randomly choosing listeners from the buffer provides diversity arising from using different checkpoints of the listener. Here the listener population was also diverse enough that helped in the learning of the speaker which would not have been the case otherwise.
> So for any task, the effect of training meta-agents will be less if we already have access to a diverse set of agents. If not, our method builds such a diverse population from scratch. Moreover, another advantage of using meta-agents is that it helps to counter adverse effects of removing agents in the buffer by reservoir sampling due to limited buffer size. So for complex tasks that require a large number of agents to be stored in the buffer, that exceeds the maximum buffer size, we would lose some listeners that could give useful signals in the training of the speaker. On the other hand, since a meta-listener would distill all past behaviors, the effect of losing one speaker would be minimal.
>
> $~$
> Moving details from Appendix to the main text
> >We will add more qualitative examples as pointed above in response to R1 & R2 and will move the section to the main paper.

---

### Official Review · Reviewer_YWdX · 2021-07-18

**Rating:** 5
**Confidence:** 4

**Summary:**

In this paper, the authors aim at training agents that can coordinate with seen, unseen/human partners in a multi-agent communication environment. Instead of using static populations, the authors propose a dynamic population-based meta-learning approach that dynamically builds the population. Such an approach enables the trained agents to generalize to seen and unseen/humans partners. On the other hand, the authors try to use a limited amount of human data to fine-tune the agents to coordinate with humans using natural language. They also compare the proposed method and baselines on two referential games.

**Ethical Concerns:**

There are human subject experimentations in Section 4.3. Do the experiments have been reviewed and approved by a relevant oversight board? I didn't find such information in the paper.


**Ethics Review Area:**

["I don’t know"]

**Limitations And Societal Impact:**

It is good that the authors could provide more details about the human experiment, such as some example conversations between the agent and human. How much does human message differ from the trained agent? What if the human is not optimal and speaks some random messages? How do you filter bad examples? Without this information, it is hard to evaluate the generalization ability of the proposed method in the human experiment.

Why does Behavior Grounding enable agents to communicate with humans using a limited amount of human data? If the human data is limited, I don't think the proposed method can generalize well on human experiments. How much does the collected human data different from the messages that appeared in the human experiments.

It is not clear how do the authors update the speaker population. Does the population get larger and larger?

The introduction of baselines is not clear as well. Which baselines corresponding to using the static population?

Figure 2 is a little bit confusing in the Dynamic Population part.

**Main Review:**

The paper is well written and easy to follow. The explanation of the problem setup and model is clear.

Different from previous works that use static populations, this paper proposed a dynamic population-based meta-learning approach. The dynamic population for multi-agent communication sounds interesting.

The experiments are good. The authors compare the proposed approach and baselines on two referential games. They also have some ablation experiments to evaluate each component of our proposed algorithm.

The authors show the results of agents interacting with humans. It is good to show the generalization ability of the proposed method. However, in this experiment, it is hard to evaluate the generalization ability of the proposed method when communicating with humans. Please see the detailed limitations below.

There is no explanation of why does Behavior Grounding enable agents to communicate with humans using a limited amount of human data? See detailed limitations below.

**Needs Ethics Review:**

Yes

**Time Spent Reviewing:**

5 hours

---

> ### Author Response · Authors · 2021-08-10
> **Response (1/2)**
>
> We thank the reviewer for their valuable feedback and address their concerns as follows:
>
> “How much does human message differ from the trained agent?”
> >The qualitative samples in Fig 12 (in the Appendix) show the generated captions for a given image for our method and each baseline including the human caption from the dataset. The BLEU scores in Table 1, 2 show the quantitative difference between the generated message and the ground-truth human utterance. We also add some corpus-level statistics for studying the linguistic diversity of the produced messages over and above the BLEU statistics already found in the paper. We will include this information in the paper.
> - Average ratio of the length of generated utterance per human utterance
> We analyze the average sentence length for all the generated utterances in the test set and compare that with the ground-truth utterances present in the dataset for the image game.
> >Our method:  $0.68 (\pm 0.05)$
> S2P:  $0.62 (\pm 0.05)$
> SIL:  $0.63 (\pm 0.05)$
> GenTrans: $0.58 (\pm 0.04)$
>
> - Average ratio of unique words in generated utterance per human utterance
> We also study the number of unique words in an utterance in the image game.
> >Our method:  $0.8 (\pm 0.03)$
> S2P:  $0.67 (\pm 0.02)$
> SIL:  $0.69 (\pm 0.02)$
> GenTrans:  $0.7 (\pm 0.03)$
>
> We would like to point out that the utterances in the dataset were collected for a different task (image captioning and machine translation). Hence the generated captions are less diverse and shorter as compared to the human captions because the underlying interactive learning task only requires “captioning in context”.
>
> $~$
> "What if the human is not optimal and speaks some random messages? How do you filter bad examples?"
> >We apologize for the missing details regarding human evaluation and we will include this information in the final version. Specifically, to filter noise in the human experiments, for each utterance we evaluated the performance of each participant against other participants. Concretely, we played each utterance of the (speaker) participant with $5$ other (listener) participants and compared the performance across all $5$ games. All utterances that do not obtain the same game score for at least $4/5$ games were excluded. In addition, additional filtering was done based on a threshold given by the BLEU score (threshold=$30$) between the participant utterance and the ground-truth utterance.

---

> > ### Comment · Reviewer_YWdX · 2021-08-24
> > **Response to Authors**
> >
> > The authors addressed some of my concerns, but some responses are not convincing enough and the paper is missing some details which I think could be improved in the next submission. I am still not sure whether the ethical response satisfying the requirement. We might need more feedback from the ethics reviewers. I will keep my original rating.

---

> > > ### Author Response · Authors · 2021-08-25
> > > **Requesting more information**
> > >
> > > We are glad to know that the response helped clarify some concerns. We tried to address each point in the review posted above.
> > >
> > > However, we are not sure which part exactly is not convincing. Could you please be more specific so that we can address any outstanding concerns?

---

> > > > ### Comment · Reviewer_YWdX · 2021-08-27
> > > > **Response to Authors**
> > > >
> > > > As the authors have agreed, many details are missing in the current submission, e.g. the details of human experiments, the details of human data, the details of the dynamic population part. Some questions are not answered, e.g. *It is not clear how do the authors update the speaker population?* and *The introduction of baselines is not clear as well.* I think this paper can be improved in the next submission, but I cannot accept it this time.
> > > >
> > > > I was also expecting to see some statistic numbers of the dataset and human experiments, such as the distribution of words, phrases, objects, etc. The authors show some examples which I think are also good, but still not convincing enough since we can always find some good examples.
> > > >
> > > > On the other hand, some responses are not convincing, i.e. "Why does Behavior Grounding enable agents to communicate with humans using a limited amount of human data?" The authors' responses do not explain why the *limited amount of human data* can do this. Maybe they think they did this, but the current response is not convincing to me.
> > > >
> > > > The ethical issue of conducting human experiments is another concern. I am still not sure whether the ethical response satisfying the requirement.

---

> > > > > ### Author Response · Authors · 2021-08-30
> > > > > **Requesting clarity on concerns**
> > > > >
> > > > > We thank the reviewer for trying to explain their concerns. However, we are not sure what part is not addressed in our responses above and so would like to better understand what exactly the reviewer means by missing details and the response not convincing.
> > > > >
> > > > > We provided all the details related to how human experiments were conducted and have also committed to adding those in the paper as asked by the reviewer and the ethics reviewers. Hence we are not sure what missing details the reviewer is referring to.
> > > > >
> > > > > We would like to point out that we have already provided statistical information about the dataset and human experiments in our first response to the review. Specifically, for our method and each baseline, we compare the length of generated utterance and the number of unique words in generated utterance with the corresponding numbers observed in the human dataset. Further, we analyze the reported numbers and provide reasoning for the observed trend.
> > > > >
> > > > > We also tried to address the confusion regarding behavior grounding and limited amount of human data. As described in our response, we wanted to clarify that the two concepts: behavior grounding and the use of limited data are orthogonal. Behavior grounding alone does not enable agents to communicate with humans using limited data. Instead, it is the combination of behavior grounding with interactive learning that increases the sample efficiency of the agents so that they are able to communicate with humans using fewer samples. This phenomenon has already been explored in much detail in the related works [23, 31, 32]. In this work, we extend this proposition and combine it with a dynamic population-based meta-learning method that helps in increasing the sample efficiency even further.
> > > > >
> > > > > Also noted in our response, L2C is a population-based method that performs meta-learning on a static population of agents. The other population-based method is Gen.Trans. which is based on cultural transmission to induce learning of compositional languages. Both the population-based methods are described at L195-206 while the single-agent methods are described at L183-194. All the baselines and their comparisons with our method are presented in Sec 3.1. The Pretrained and EmeCom baselines are defined at L240 and L245 respectively.
> > > > >
> > > > > Regarding updating the population, we noted in our response that each population (for both speaker and listener) grows gradually as we add new agents at every iteration. Fig 2 and its caption along with Fig 8 in the Appendix also depict this iterative process. We divide the whole dynamic population update into two phases. First, a meta-listener is trained using the current population of speakers at time t. Then a new speaker, initialized with the parameters from the latest speaker obtained at t−1,  is trained with the updated meta-listener and added to the speaker population at time t.

---

> ### Author Response · Authors · 2021-08-10
> **Response (2/2)**
>
> "Why does Behavior Grounding enable agents to communicate with humans using a limited amount of human data?"
> >Behavior Grounding essentially provides the grounding signal during training such that the language learnt is compatible with the human prior. Since the ultimate goal is to communicate with humans, it is necessary to provide information about human bias to enable the agents to find a protocol that is consistent with both the given task and human distribution.
> In general, behavior grounding can require more or less human samples for good performance. Indeed, previous work [23, 31, 32] has shown that we can achieve better sample efficiency when behavior grounding is combined with interactive learning. We extend this idea in a dynamic population-based setup which allows us to improve the sample efficiency even further.
>
> "How much does the collected human data different from the messages that appeared in the human experiments."
> >As noted in the previous answer, we compute a similarity (BLEU) score between the data collected by the human experiments and the utterances found in the dataset. We found that all participant utterances gave a BLEU score in the range $30-35$. Furthermore, we show some actual samples collected during human experiments along with their dataset utterance (for the image game) and will include them in the final version.
>
> - Dataset: `one person sitting with a drink and one flying a kite on the beach.`
> Human experiments:
> `A person flying a kite on a beach with another person sitting.`
> `A man flying a colored kite and a person sitting with a drink at a beach.`
> `A person flying a kite while another person sitting with a drink on a beach.`
> $~$
> - Dataset: `a park filled with different color umbrellas hanging from trees..`
> Human experiments:
> `Many colored umbrellas hanging on trees.`
> `Colored umbrellas hanging on trees in a forest.`
> `A lot of colored umbrellas hanging on different trees.`
>
> $~$
> "Does the population get larger and larger?"
> >Yes, the population (in the buffer) grows gradually as new agents are added at every iteration as described in the paragraph at L158 and Fig 2b and a detailed illustration in Fig 8. We used a buffer size of $200$ each for storing speaker and listener parameters as noted in L219.
>
> $~$
> "Which baselines corresponding to using the static population?"
> >L2C [30] uses a static population of agents as noted in L37, 197. We will also make the description of baselines clearer by separating population-based and the rest of the baselines.
>
> $~$
> Ethical Concerns
> > We have addressed this in the common response to Ethics Reviewers above.

---

### Official Review · Reviewer_ctGD · 2021-07-21

**Rating:** 7
**Confidence:** 4

**Summary:**

The current work proposes to use dynamic population based meta-learning in referential games between agents using natural language. In particular, it uses meta-learning to capture the diverse speaker/listener behaviors in a population and then initialize new agents by interacting with these meta-agents across iterations.  They also add supervised learning iterations to ensure there is no language drift. Experiments show that agents trained through the proposed approach have higher accuracy in the referential game interacting within themselves and also with humans, while retaining the naturalness of the language.

**Limitations And Societal Impact:**

The authors briefly discuss the limitations of the proposed approach but do not discuss broader impact in the main paper.

**Main Review:**

**Strengths**
(S1) The idea to use a meta-listener and meta-speaker across different iterations in dynamic populations is indeed an interesting and novel direction. It is also well motivated in the paper.

(S2) Experimental evaluation is solid and backs up the claims in the earlier part of the text. The increase in referential game accuracy (with statistical significance) and improved BLEU score when compared to ground truth captions both evidence the effectiveness of the proposed solution. The human studies are the cherry on top where the human-agent teams trained using the proposed approach outperform other existing approaches.

(S3) The authors did a good job writing the manuscript as it is easy to read and understand. The work is well placed in the context of related work and all the experimental details are clearly explained.

**Weaknesses**
(W1) The manuscript does not contain any qualitative examples in the main paper. Some have been included in the appendix. Further, it would be interesting to see where the agents fail to better understand the strengths and weaknesses of the proposed approach. For instance, categorizing the errors due to imperfect vision, incorrect language usage by the speaker, or language understanding by the listener would be useful.

**Comments**
(C1) How important is the role of supervised learning iterations in order to retain the naturalness of the language? The current work is missing this ablation, where the amount of supervised learning is varied.

(C2) To check for robustness against any implicit bias in image datasets, how do agents trained on MSCOCO perform on a different image dataset at test time, e.g, PASCAL?

**Typos**
* Algorithm 1: After the meta learning stage, is there a typo in the initialization of \phi’_0?
* L15: your -> our


**Time Spent Reviewing:**

2

---

> ### Author Response · Authors · 2021-08-10
> **Response (1/2)**
>
> We appreciate the positive feedback by the reviewer and address their concerns as follows:
>
> W1:
> As the reviewer notes, we show some qualitative samples in the Appendix and will move the section with some more examples in the main paper. We welcome the reviewer’s suggestion and show some failure examples of our model under different categories. We will include more examples with images in the paper and highlight any further types of error arising from our agent.
>
> - Incorrect message due to imperfect vision
> >Dataset:
>     `a big black bear that is walking into the road`
> Speaker message:
>     `A black dog crossing the road with cars.`
> (In this test example, the speaker incorrectly uses dog for a bear.)
>
> - Incorrect language usage by the speaker
> >Dataset:
>     `a young man with a lacrosse stick and nike bag dressed in a shirt and tie.`
> Speaker message:
>     `A group of people standing with bags, tie, and bottle.`
> (Here, the speaker tried to add all observed objects in the message without being semantically correct.)
>
> - Incorrect language understanding by the listener
> >Speaker message:
>     `A child with a teddy bear.`
> Listener chooses an incorrect target image that only contains teddy bears confusing the teddy bear with a child.
>
> From the analysis so far, we believe that the speaker gets confused when it encounters an out-of-vocabulary object in the target image. Likewise, the listener chooses a wrong target image when the complexity of the distractors is increased (by using distractors with similar objects as in the target image). Nonetheless, we think these issues are not specific to our agent and would arise in any interactive learning model as we scale up to include more and more objects.

---

> ### Author Response · Authors · 2021-08-10
> **Response (2/2)**
>
> C1:
> The supervised learning phase is important to the learning of the agents to keep them grounded in the human distribution when they are additionally asked to perform well on a given task. Previous work [23, 31, 32] has investigated this question in detail in the context of language drift and provided empirical evidence that supports the hypothesis, hence we didn’t include an ablation without any supervised learning. However, we do touch on questions on what leads to better language performance, e.g.,
> 1) Even in our experiments, L2C (which only uses supervised learning after initialization) performs much worse on both the referential accuracy (Sec 4.1) and naturalness of the language (Sec 4.2).
> 2) Our ablation “no separate grounding” utilizes a KL divergence loss, i.e. we conduct grounding in a different way rather than using a separate supervised learning loop. Despite the use of a  different grounding method, we show that our method performs almost the same, suggesting that the way the grounding signal is conveyed to our agents has little effect on their overall performance, also noted at L366-368.
>
> C2:
> We thank the reviewer for this suggestion and will include this in the final version.
> We ran experiments on the Flickr8k dataset [Hodosh et al. 2013] with $5000$ train images and $1000$ val/test images. We conduct a cross-task evaluation experiment on the referential game with $K=9$ distractors where we evaluate on Flickr8k but train on MSCOCO and a within-task evaluation where we train and evaluate on Flickr8k. We evaluate our model and the Pretrained baseline on the referential accuracy.
>
> Test on Flickr8k:
> Ours
> >Cross-task (trained on MSCOCO): $70.2 (\pm 1.8)$
> Within-task (trained on Flickr8k): $78.9 (\pm 1.4)$
>
> Pre-trained
> >Cross-task (trained on MSCOCO): $59.7 (\pm 0.9)$
> Within-task (trained on Flickr8k): $65.3 (\pm 0.3)$
>
> While we do observe a drop in the performance when compared to the within-task performance ($70.2$ v/s $78.9$), our model still outperforms the Pre-trained baseline ($70.2$ v/s $65.3$). This observation correlates with our results using human players and out-of-population evaluation show in Sec 4.3 and 4.5 respectively.
>
> - Hodosh et al. Framing Image Description as a Ranking Task: Data, Models and Evaluation Metrics. JAIR 2013.
>
>
> $~$
> Thanks for pointing out the typos, we will fix them in the final version.

---

### Review · Ethics_Reviewer_b7jJ · 2021-08-10

**Recommendation:**

Given that the conference expects studies involving human subjects to be reviewed and approved by relevant oversight boards, it would be nice if the authors can provide such information. And also, even beyond the purpose of IRB approval, information about the

I encourage authors to provide information on the human subjects including if they were compensated for taking part in the experiment as well as some general relevant background information  -- as this is a factor that likely impacts the performance of the agents (section 4.3 Agents Interacting with Humans). Inclusion of such in-depth discussion would make the paper richer.


**Ethical Issues:**

Yes

**Ethics Review:**

This work involves human subjects. However, the authors do not provide information on whether the experimentation was reviewed and approved by the relevant oversight board. Neither do they provide information involving the humans in the experiment.

---

> ### Author Response · Authors · 2021-08-17
> **Response**
>
> We thank the reviewer for their suggestions.
>
> We have addressed all concerns in the common response above.

---

### Review · Ethics_Reviewer_vehr · 2021-08-11

**Recommendation:**

The authors should include more information about any human experiments that they ran: how they experiments were run, any compensation details they can offer, and any potential risks to participants. If the authors revise their paper to include these, it will address this ethical question.

**Ethical Issues:**

Yes

**Ethics Review:**

Reviewer YWdX mentioned that this paper uses the results of human experiments (especially in Section 4.3) but is not very transparent about how these experiments were run. I’m a bit confused by the authors’ response to Section 5 of the Checklist: in 5a, they state that they included the full text of instructions given to participants, in 5b, that they described any participant risks, and in 5c they state that the issue of compensation isn’t relevant. However, I looked through the main body of the text and the appendix and didn’t seem to find anything that would match the content in 5a or 5b, and I do believe that 5c is relevant (even if the participants aren’t compensated, that should be described).

This might be a misunderstanding about what exactly we’re looking for. One useful model paper may be https://arxiv.org/pdf/1802.07810.pdf, which also uses human experiments. In Appendix B, they include full text copies of instructions for participants. This may be long, but if it is in the appendix, that shouldn’t matter. On page 10 they state the average compensation for participants and also note that their experimental set-up passed an internal Microsoft review (items 5b and 5c on the checklist). Including notes similar to those (or potentially more detailed would be even better) would help to address these ethics issues.

---

> ### Author Response · Authors · 2021-08-17
> **Response**
>
> We thank the reviewer for pointing out the missing information.
>
> We have addressed each point 5a, 5b, 5c of the checklist in the common response above.

---

### Author Response · Authors · 2021-08-17
**Common response to Ethics Reviewers**

We thank the ethics reviewers for their valuable comments and apologize for the missing/unclear information.

We would like to note that we do provide information about the number of participants used for each experiment and the total number of participants used and their affiliation on L623-642 in the Appendix and Sec 4.3.
Our human experiments were done in a controlled environment with a group of 45 undergraduate and graduate students studying various arts and science disciplines at our university. The experiments were overseen and approved by our internal lab/university review board. The participants weren’t compensated for taking part in the experiments as our lab/university has been conducting such experiments in the past on a quid pro quo basis.
Moreover, the participants were given the following instructions to play the game and were ensured that their individual identities would not be revealed or used in a way that could contaminate our results. Consequently, there were no participant risks involved in our experiments.

Overall instructions:
>We are interested in conducting human experiments for the popular referential games proposed in Lewis et al, ‘69. It is a cooperative game that involves two players: a speaker and a listener. A speaker observes a target image and emits a message that is sent to the listener. The listener looks at the message and tries to identify the target image among a set of distractor images. Both agents are given a positive reward if the listener’s prediction is accurate and zero otherwise.

>Our experiments require each participant to play as a speaker and a listener with different partners. Your partner could be a human or one of our trained agents. You will not be given the identities of your partners and your individual identities will not be used for analyzing the final results.

Message to the human speaker:
>You are assigned the role of a speaker.
Look at the image carefully and write a caption that best describes the image.

Message to the human listener:
>You are assigned the role of a listener.
Read the message carefully and use that to choose the target image for which the message was intended, among the set of other distractor images.

We will include some actual examples in the final version along with the images that were shown to the participants.

As noted in footnote 2, all participants were paired with randomly chosen partners in consecutive games to not allow them to form ad-hoc conceptions which is a common pitfall of such experiments and hence crucial to the experimental setup.

Finally, we would like to highlight that revealing further information about the participants could lead to disclosing information about our affiliations and hence deanonymizing the author identities, which is also why we chose not to include it in the submission. Nevertheless, we will include the above information in the final version and describe the human evaluation in more detail.

---

### Decision · Program_Chairs · 2021-09-27

**Decision:**

Accept (Poster)

**Comment:**

This paper introduces an interesting approach to combating semantic drift in communication-based training for language agents. The biggest concern on reviewers was the need to more clearly describe the details of methods and results for the human experiments. It will be very important to add methods, ethics, and results details in the revision. Some reviewers felt that the domains considered were too artificial, but I believe (along with other reviewers) that these are an appropriate first step given previous work in this area.